# Arf1/COPI machinery acts directly on lipid droplets and enables their connection to the ER for protein targeting

Florian Wilfling[1†], Abdou Rachid Thiam[1,2†], Maria-Jesus Olarte[1], Jing Wang[1], Rainer Beck[1,3], Travis J Gould[1], Edward S Allgeyer[1], Frederic Pincet[1,2], Jörg Bewersdorf[1], Robert V Farese Jr[4,5,6]*, Tobias C Walther[1]*

[1]Department of Cell Biology, Yale University School of Medicine, New Haven, United States; [2]Laboratoire de Physique Statistique UMR 8550, Ecole Normale Supérieure de Paris, Université Pierre et Marie Curie, Université Paris Diderot, Centre National de la Recherche Scientifique, Paris, France; [3]Heidelberg University Biochemistry Centre, University of Heidelberg, Heidelberg, Germany; [4]Gladstone Institute of Cardiovascular Disease, San Francisco, United States; [5]Department of Medicine, University of California, San Francisco, San Francisco, United States; [6]Department of Biochemistry and Biophysics, University of California, San Francisco, San Francisco, United States

**Abstract** Lipid droplets (LDs) are ubiquitous organelles that store neutral lipids, such as triacylglycerol (TG), as reservoirs of metabolic energy and membrane precursors. The Arf1/COPI protein machinery, known for its role in vesicle trafficking, regulates LD morphology, targeting of specific proteins to LDs and lipolysis through unclear mechanisms. Recent evidence shows that Arf1/COPI can bud nano-LDs (~60 nm diameter) from phospholipid-covered oil/water interfaces in vitro. We show that Arf1/COPI proteins localize to cellular LDs, are sufficient to bud nano-LDs from cellular LDs, and are required for targeting specific TG-synthesis enzymes to LD surfaces. Cells lacking Arf1/COPI function have increased amounts of phospholipids on LDs, resulting in decreased LD surface tension and impairment to form bridges to the ER. Our findings uncover a function for Arf1/COPI proteins at LDs and suggest a model in which Arf1/COPI machinery acts to control ER-LD connections for localization of key enzymes of TG storage and catabolism.

*For correspondence: bfarese@ gladstone.ucsf.edu (RVF); tobias. walther@yale.edu (TCW)

†These authors contributed equally to this work

## Introduction

Nearly all organisms balance fluctuations in the availability of energy sources with the need for energy expenditure. With its high energy content, triacylglycerol (TG) stored in lipid droplets (LDs) is the primary means of storing energy for many organisms (*Thiele and Spandl, 2008*; *Fujimoto and Parton, 2011*; *Brasaemle and Wolins, 2012*; *Walther and Farese, 2012*). LDs also store lipid precursors for membrane synthesis (e.g., cholesterol and glycerophospholipids) needed, for example, when cells exit quiescence and expand membranes for cell division (*Kurat et al., 2009*). Due to their function in lipid storage, LDs are central to the development of pathologies associated with excess lipid accumulation, ranging from atherosclerosis and cardiovascular disease to obesity and metabolic syndrome (*Krahmer et al., 2013*).

Unlike most organelles, LDs are not delimited by a bilayer membrane but instead are covered with a monolayer of phospholipid surfactant, which is important for their stability in cells (*Tauchi-Sato et al., 2002*; *Krahmer et al., 2011*; *Yang et al., 2012a*). In this sense, LDs constitute the dispersed phase of a cellular emulsion, with the phospholipid monolayer acting as a surfactant at the interface of the oil core with the aqueous cytosol (for review, see *Thiam et al., 2013b*). Proteins specifically located at the LD

**eLife digest** Just like the body contains organs that perform different jobs, the cells within the body contain organelles that carry out different tasks. The endoplasmic reticulum, for example, makes proteins that are sent to other organelles or to destinations outside the cell. Each organelle is typically sealed within a membrane made from a double layer of phospholipids—molecules that have a phosphate 'head' group at one end, and two fatty acid 'tails' at the other. Proteins are shuttled between the organelles inside membrane-bound packages called vesicles.

There is, however, an exception to this rule. Lipid droplets are organelles that store fats and oils inside a single layer of phospholipids. This layer can include enzymes that break down the contents of the droplet, or make new fat molecules, depending on the needs of the cell and the organism. However, it is not clear how these enzymes get from the endoplasmic reticulum to the lipid droplet.

Previous work had suggested that a protein complex called Arf1/COP—which is also involved in the movement of vesicles around the cell—might recruit the enzymes to the lipid droplets. However, none of the other proteins known to be involved in vesicle transport were needed to transport the enzymes to the droplets, which suggested that the Arf1/COPI complex was using a previously unknown mechanism to move the enzymes.

Now Wilfling, Thiam et al. have shown that Arf1/COPI complexes trigger the establishment of membrane bridges between the endoplasmic reticulum and the droplets, which means that vesicles are not needed to get the enyzmes to the lipid droplets. It was also shown that the Arf1/COPI complexes could pinch off tiny droplets from full-size lipid droplets taken from living cells. Wilfling, Thiam et al. suggest that this 'budding' process changes the composition of the phospholipid layer around the larger droplet in a way that allows it to interact directly with the membrane of the endoplasmic reticulum.

By providing new insights into the trafficking of proteins between organelles, the work of Wilfling, Thiam et al. reveals mechanisms that govern the composition of lipid droplets. In the future, these pathways could be manipulated to treat conditions that result from excessive storage of fat, such as obesity or cardiovascular diseases.

surface execute many of the reactions of lipid storage or mobilization. For example, enzymes mediating TG synthesis and hydrolysis localize to LDs, where they mediate LD expansion and shrinkage, respectively (*Kuerschner et al., 2008*; *Schweiger et al., 2008*; *Stone et al., 2009*; *Murugesan et al., 2013*; *Wilfling et al., 2013*). How such enzymes are specifically targeted to LDs is a poorly understood, yet fundamental question.

Unbiased genome-wide screens in model systems, such as *Drosophila* cells, revealed factors that are required for LD targeting of proteins (*Beller et al., 2008*; *Guo et al., 2008*). Specifically, members of the Arf1/COPI machinery, but not other proteins involved in secretory trafficking (e.g., COPII or clathrin), are necessary for normal LD morphology and for the targeting of some proteins to LDs (*Beller et al., 2008*; *Guo et al., 2008*; *Soni et al., 2009*). Depletion of Arf1/COPI proteins from cells leads to the formation of relatively uniform LDs of a characteristic size that exhibit impaired lipolysis (*Beller et al., 2008*; *Guo et al., 2008*). Consistent with this, Arf1/COPI proteins are required for LD localization of the major TG lipase ATGL (*brummer* in *Drosophila*) (*Beller et al., 2008*; *Soni et al., 2009*; *Ellong et al., 2011*). ATGL was shown to behave, biochemically, as an integral membrane protein (*Soni et al., 2009*), and it was suggested that this lipase is transported to LDs from the ER by vesicular trafficking.

In vesicular trafficking, the best-characterized function of Arf1/COPI proteins is in retrograde transport, that is, retrieving ER resident proteins from the Golgi apparatus (*Nickel et al., 2002*). In this pathway, Arf1 is loaded with GTP by a nucleotide exchange factor, such as GBF1 [*gartenzwerg (garz)* in *Drosophila*]. The activated Arf1-GTP then recruits the coatomer, a heptameric protein complex, leading to the formation of a coated transport vesicle. Subsequent uncoating of the vesicle allows its transport and fusion to the target membrane (e.g., the ER).

It is unknown how Arf1/COPI proteins function in LD biology. Although one possibility is that Arf1/COPI proteins target proteins to LDs via bilayer vesicles, a variety of studies suggest a function directly at LDs. First, Arf1 and its GEF, GBF1, as well as other members of the COPI machinery, have been found

on LDs in proteomic and cell biological studies (*Nakamura et al., 2005*; *Bartz et al., 2007*; *Ellong et al., 2011*; *Bouvet et al., 2013*). Second, the expression of dominant-negative Arf1T31N, which binds its exchange factor tightly, localizes to LDs (*Guo et al., 2008*). Third, Arf1Q71L that cannot hydrolyze GTP (and hence acts as a dominant-negative mutant in vesicular trafficking) activates lipolysis from LDs (*Guo et al., 2008*). Most recently, GTP-bound Arf1 and COPI proteins were shown to bud nano-LDs of ~60 nm diameter from a phospholipid covered oil-water interface in vitro (*Thiam et al., 2013a*), indicating that this machinery can interact with monolayer interfaces such as what is found at LD surfaces. Collectively, these data suggest an alternative, so far untested model, in which Arf1/COPI proteins function in cells directly at LDs in a way that enables protein targeting.

Besides ATGL, other enzymes involved in TG metabolism also localize to LD surfaces. For example, at least one isoenzyme catalyzing each step of de novo TG synthesis from glycerol-3-phosphate (e.g., GPAT4, AGPAT3, and DGAT2) localizes to a subset of LDs. Each of these enzymes contains two hydrophobic segments likely forming a hairpin in the ER membrane or the LD monolayer (*Wilfling et al., 2013*). LD localization of these enzymes enables LDs to synthesize TG locally and expand their neutral lipid cores under conditions of excess energy (fatty acid) availability. Recent evidence indicates that these enzymes re-localize to a subset of LDs from the ER via abundant membrane bridges that form between the organelles (*Wilfling et al., 2013*; *Thiam et al., 2013a*). Intriguingly, this targeting reaction can occur rapidly at a particular LD, from which TG synthesis enzymes were absent for a long time (*Wilfling et al., 2013*). How the targeting process is initiated and how bridges between LDs and the ER are established is unknown.

Here we investigate the mechanism of Arf1/COP protein function in cellular LD protein targeting by using a combination of cell biological and biochemical approaches. In contrast to the canonical role of these proteins in vesicular trafficking, we uncover a mechanism of action that relies on altering the surface lipid composition of LDs. Based on the presence of the Arf1/COPI machinery at LDs, we propose a newly identified function of Arf1/COPI proteins in modulating LD surfaces to enable protein targeting.

## Results

### Arf1/COPI are required for lipid droplet targeting of triglyceride synthesis enzymes

Many cell types, including *Drosophila* S2 cells, contain two populations of LDs: a few rather large, expanding LDs, several microns in diameter, and many smaller (less than a micron diameter) LDs (*Wilfling et al., 2013*). Depletion of the *Drosophila* Arf1 homologue Arf79F or βCOP results in a relatively uniform LD population (*Beller et al., 2008*; *Guo et al., 2008*). We quantified this phenotype and found that depletion of either Arf79F or βCOP results in a relatively narrow, monodisperse distribution of LDs that lies intermediate in size (mean ~1.3 μm) between small and larger expanding LDs (*Figure 1A*).

Since the Arf1/COPI-depleted cells lacked large expanding LDs, we tested whether Arf1/COPI depletion affected the LD localization of enzymes catalyzing LD expansion by examining LD localization of fluorescent GFP-tagged or endogenous GPAT4 (detected by immunofluorescence). We found that depletion of Arf79F, *garz*, or any of the coatomer subunits, with the exception of εCOP, impaired the LD localization of GPAT4 (*Figure 1B*, *Figure 1—figure supplement 1B,C*). Similarly, depletion of Arf79F or βCOP compromised LD targeting of the triglyceride-synthesis enzyme DGAT2 (*Figure 1— figure supplement 1D*). Defective GPAT4 localization to LDs with Arf1/COPI depletion was also evident in subcellular fractionation experiments, where the amount of GPAT4 in the LD fraction was greatly diminished in the absence of βCOP (*Figure 1C*). Consistent with previous reports (*Beller et al., 2008*; *Soni et al., 2009*), we found that *brummer* was also missing from LDs in Arf1/COPI-depleted cells (*Figure 1— figure supplement 1E*). The targeting defect was apparently specific to proteins targeting LDs from bilayer membranes, as at least some proteins that localize to LDs from the cytoplasm, such as the *Drosophila* perilipin Lsd1, were not affected by Arf1/COPI depletion (*Figure 1D*). The absence of TG synthesis enzymes likely explains the absence of large LDs, and the defect in lipase targeting and the associated defect in lipolysis, likely contribute to the increase in size of small LDs to an intermediate size.

### Arf1/COPI proteins trigger the formation of LD–ER membrane bridges, enabling rapid protein targeting to LDs

Some of the proteins requiring Arf1/COPI for LD localization, such as specific isoenzymes of TG synthesis (including GPAT4), access LDs from the ER through membrane bridges (*Wilfling et al.,*

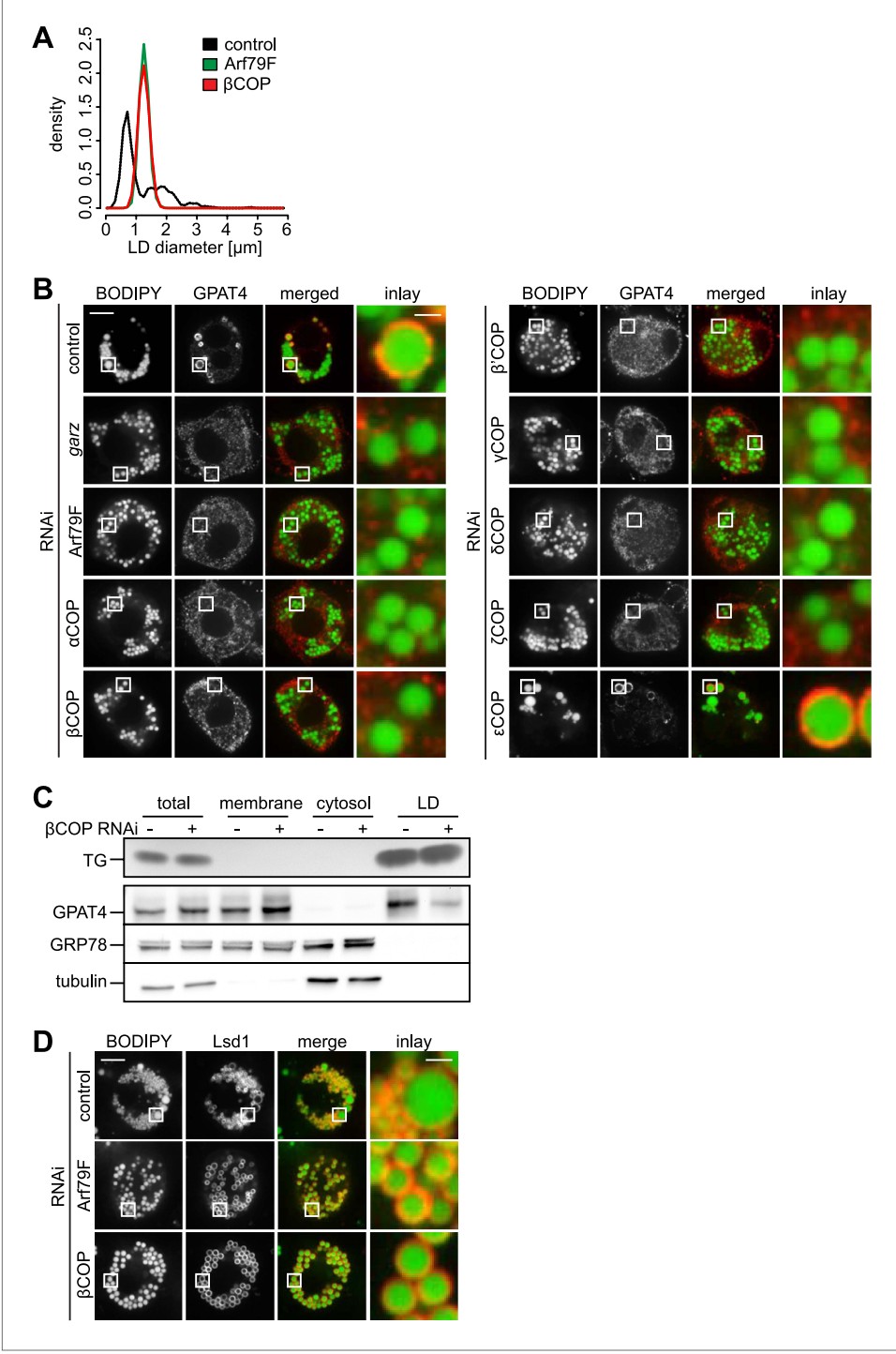

**Figure 1**. The COPI machinery is required for LD targeting of specific proteins. (**A**) The bimodal size distribution of control cells (black line) with few large LDs and many small LDs shifts a monodisperse size in Arf1/COPI-depleted cells (green and red line). The figure shows the density function of the LD size distribution. (**B**) Endogenous GPAT4 detected by immunofluorescence localizes to LDs (stained by BODIPY) in control treated cells, but not in the absence of any of the COPI machinery components, except εCOP. (**C**) The amount of GPAT4 fractionating with LDs (detected by thin layer chromatography of TG) is reduced in cells depleted of βCOP. (**D**) Arf1/COPI effects on LD protein targeting are protein specific, as Lsd1 targeting to LDs is not affected in cells depleted of Arf1/COPI. *Cherry*-Lsd1 localizes to LDs stained with BODIPY in the absence of Arf79F (middle panel) or βCOP (bottom panel). Scale bars are 10 μm (overview) or 1 μm (inlay).

*Figure 1. Continued on next page*

*Figure 1. Continued*

The following figure supplements are available for figure 1:

**Figure supplement 1**. The COPI machinery is required for LD targeting of specific proteins.

*2013*). We hypothesized that Arf1/COPI activity on LDs is required for establishing the junctions between the ER and LDs.

To test this hypothesis, we performed add-back experiments with Arf1/COPI in GPAT4 localization assays. We fused LD-containing cells depleted for βCOP and expressing GFP-tagged GPAT4 localized in the ER, with wild-type cells that provide Arf1/COPI proteins in trans (*Figure 2A*). After cell–cell fusion, the COPI pool from wild-type cells rapidly equilibrates in the mixed cytoplasm. This led to rapid targeting of GFP-GPAT4 to some of the pre-existing LDs (*Figure 2B*), with a variable lag phase of 1–25 min (*Figure 2C*). LD targeting of GPAT4 invariably occurred directly from the ER through a number of

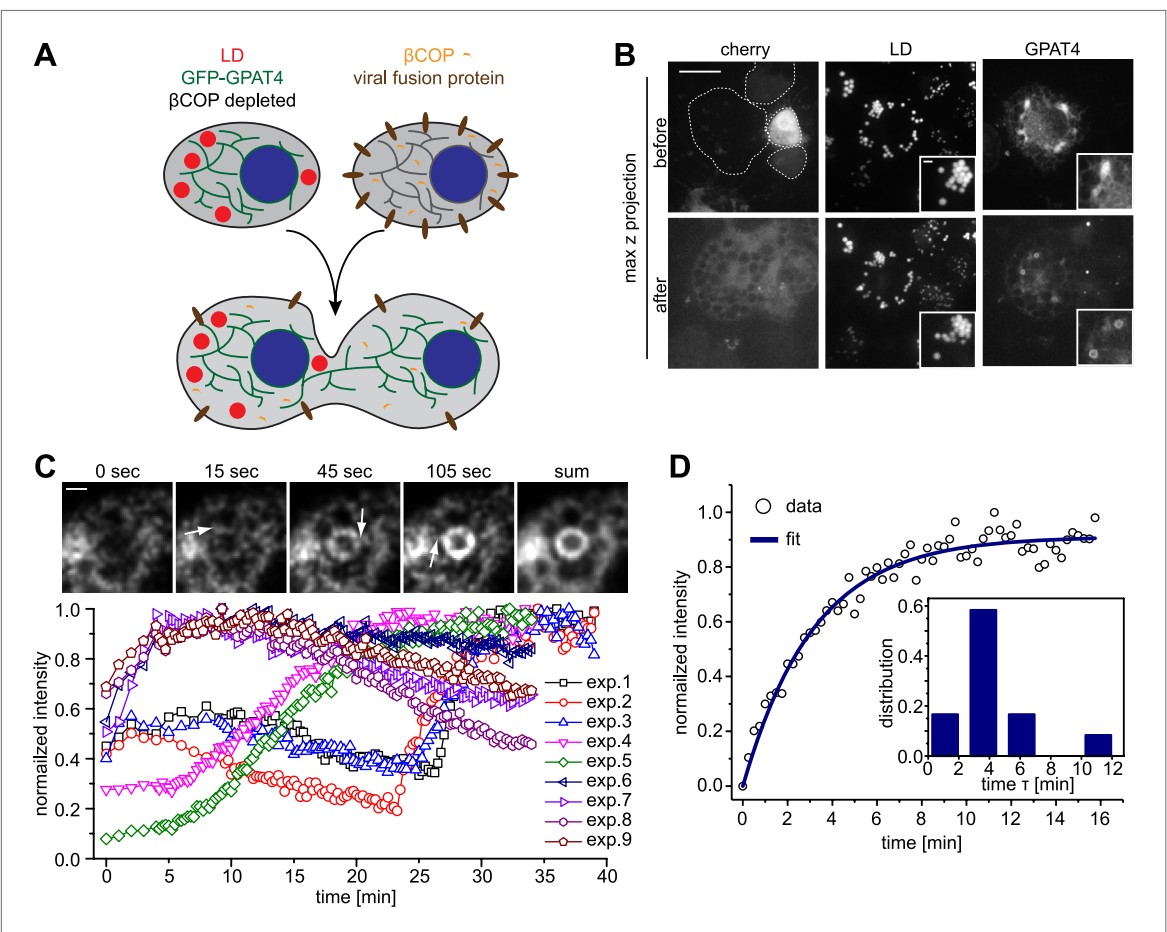

**Figure 2**. Arf1/COPI mediate LD protein targeting by establishing connections between the ER and LDs. (**A**) Schematic representation of cell–cell fusion experiments. (**B**) Fusion of βCOP depleted cells expressing GFP-GPAT4 and induced LDs with WT cells rapidly rescues GFP-GPAT4 targeting to LDs. Soluble cherry fluorescent protein is expressed as a marker for content mixing of fused cells. Scale bars are 10 μm (overview) or 1 μm (inlay). (**C**) Time lapse analysis of GFP-GPAT4 targeting to LDs. Upper panels shows representative images from time-lapse imaging of cell–cell fusion experiments. Arrows point to apparent connections between LDs and the ER. Scale bar = 1 μm. Lower panel shows quantitation of GFP-GPAT4 localization to LDs in nine independent cell–cell fusion experiments. Time = 0 indicates fusion and content mixing of cells. (**D**) Rate of GFP-GPAT4 recruitment to LDs after cell–cell fusion. Insert shows the histogram of characteristic recovery time τ.

The following figure supplements are available for figure 2:

**Figure supplement 1**. Mathematical model for GPAT4 targeting to LDs through bridges with the ER.

junctions between the two organelles (*Figure 2C*; *Video 1*). After the initial lag phase, GPAT4 targeting was rapid, with a characteristic time τ of 3.6 ± 1 min (*Figure 2D*, *Figure 2—figure supplement 1A–C*). A mathematical model using the (experimentally determined) apparent diffusion constant of GPAT4 in the ER (0.035 ± 0.005 µm$^2$/sec) revealed that roughly 5–9 connections between a LD and the ER are required to obtain the observed speed of GFP-GPAT4 targeting to LDs (*Figure 2—figure supplement 1B*, 'Materials and methods'). This is consistent with the observed number of connections to large LDs in fluorescence and EM images (*Figure 2C*, FW, MJO, and TCW, unpublished observations).

## Arf1/COPI proteins localize to the lipid droplet surface

We next asked how Arf1/COPI proteins trigger the formation of LD-ER connections. If Arf1/COPI proteins act directly on LDs in this process, then a portion of these proteins should localize to LDs. To test this, we determined the localizations of the Arf1 exchange factor *garz* and αCOP in *Drosophila* S2 cells. For each protein, we observed foci localizing to the surface of some LDs in addition to signal likely reflecting the Golgi pool of the proteins (*Figure 3A*). Importantly, LD co-localization occurred more frequently than would be expected by overlaying a random pattern of foci onto the LD signals (*Figure 3A*, *Figure 3—figure supplement 1A*). We also observed abundant co-localization of GFP-tagged Arf79F with the LD marker CGI-58 on LDs (*Figure 3—figure supplement 1B*), and the signal was distinct from signals marking the Golgi apparatus (*Figure 3—figure supplement 1C*).

To test whether Arf1/COPI localization to LDs was conserved between different species, we localized GBF1, β'COP, and βCOP in mammalian NRK cells. Similar to findings in *Drosophila* cells, some signal from each protein localized in foci to LDs (*Figure 3B*, *Figure 3—figure supplement 1E*). Also in this case, LD colocalization of the coatomer protein βCOP was overrepresented, compared with the expectation for a random pattern. Moreover, βCOP and β'COP were localized to the same foci (*Figure 3—figure supplement 1F*). In contrast, clathrin or KDEL receptor were either underrepresented or only occasionally showed a focus on LDs, consistent with a randomly distributed pattern (*Figure 3B*, *Figure 3—figure supplement 1G*). In NRK cells, the colocalizing foci completely overlapped with the LD marker perilipin3 in confocal and super-resolution stimulated emission depletion (STED) images (*Figure 3C*, *Figure 3—figure supplement 1H*), but were distinct from signals marking the Golgi apparatus (GM130, *Figure 3—figure supplement 1I*). As expected by its interaction with GTP-loaded Arf1, LD localization of COPI coat was blocked completely when cells were incubated with brefeldin A (*Figure 3D*, *Figure 3—figure supplement 1D*), which inhibits Arf1 nucleotide exchange factors. This inhibition by brefeldin A leads to the formation of a stable, abortive complex of the compound with Arf1 (*Walker et al., 2011*). Similar results were obtained with Golgicide A, a specific inhibitor of GBF1 Arf1 exchange factors (*Figure 3D*; *Dobrosotskaya et al., 2002*).

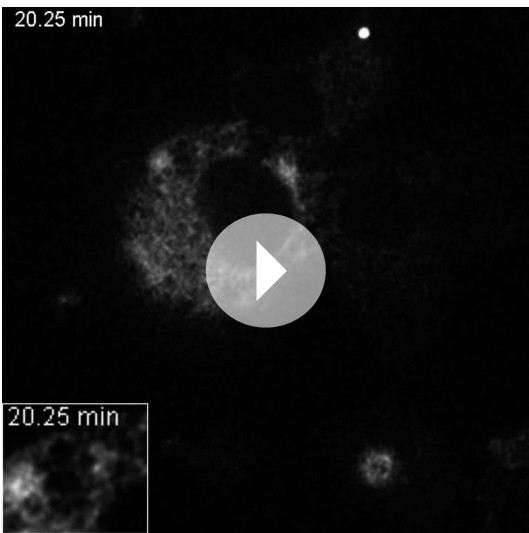

20.25 min

20.25 min

**Video 1**. Time lapse analysis of GFP-GPAT4 targeting to LDs in cell–cell fusion experiments.

## Arf1/COPI proteins bud nano-lipid droplets from existing lipid droplets

The presence of Arf1/COPI on LDs prompted us to test whether this machinery can bud nano-LDs from cellular LDs, similar to the way it forms COPI-coated nano-LDs from artificial oil-water interfaces (*Thiam et al., 2013a*). We isolated LDs from *Drosophila* S2 cells and incubated them with purified Arf1/COPI proteins. Electron microscopy revealed that specifically in the presence of the Arf1/COPI machinery and a non-hydrolyzable GTP analogue (GTPγS), abundant protein-covered nano-LDs were formed (*Figure 4A*). The nano-LDs had an average diameter of 65 nm ± 10 nm (*Figure 4A*), consistent with the size range of COPI-coated vesicles (*Simon et al., 1996*) or the size of COPI-coated nano-LDs formed at artificial oil-water interfaces (*Thiam et al., 2013a*). For vesicle formation by COPI, dimerization of Arf1 is required (*Beck et al., 2008*). Interestingly, the formation of nano-LDs was unaffected in reactions

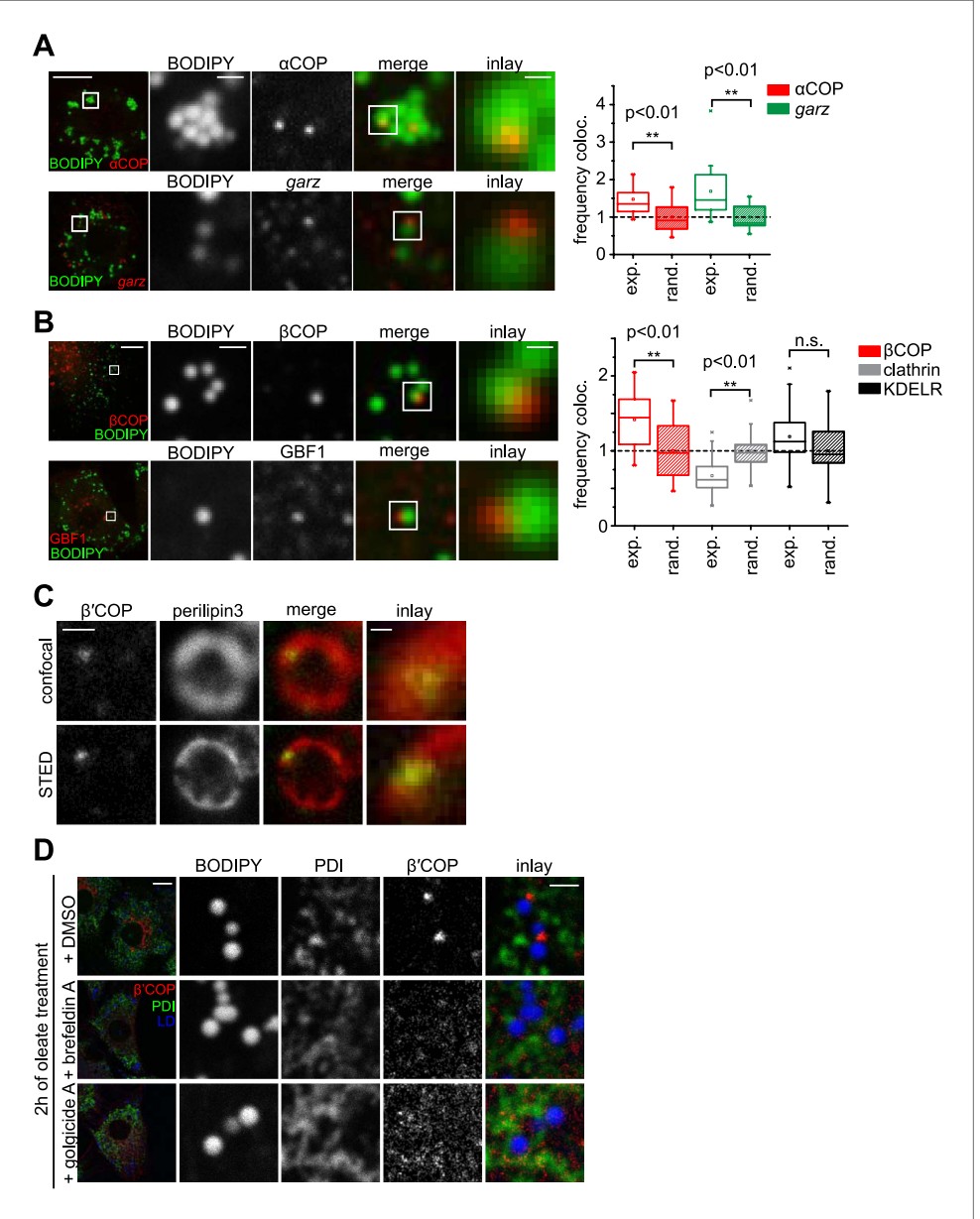

**Figure 3**. The COPI machinery localizes to the LD surface. (**A**) The endogenous COPI machinery stained with αCOP or *garz* antibodies (red) localizes to LDs in S2 cells. Frequencies of colocalization of αCOP and *garz* spots with LDs from experiments are higher than expected from a random distribution. (**B**) The endogenous COPI machinery localizes to LDs in NRK cells. NRK cells stained for βCOP or GBF1 by immunofluorescence (red) show partial colocalization with LDs stained with BODIPY (green). Colocalization of βCOP with LDs in NRK cells is not random. Relative frequencies of βCOP, KDEL receptor and clathrin spots colocalizing with LDs determined in experiments are respectively compared to the frequencies of colocalization from a binomial random distribution. From the two frequencies (experiment vs simulation), a significant overrepresentation of βCOP on LDs is observed, whereas clathrin and KDEL receptor (KDELR) are not found on LDs. For (**A**) and (**B**) scale bars are 10 μm (overview) or 1 μm (first inlay) or 250 nm (second inlay). Statistical significance was tested by a student *t* test with p<0.01 (n = 30). (**C**) Localization of β'COP (green) to the LD surface (perilipin3, red) using confocal (upper panel) and super-resolution STED microscopy (lower panel). Scale bar = 500 nm (overview) or 100 nm (inlay). (**D**) Localization of β'COP to LDs is efficiently blocked by treatment of cells with the Arf1 GEF inhibitors brefeldin A or golgicide **A**. Scale bar = 10 μm (overview) or 1 μm (inlay).

The following figure supplements are available for figure 3:

**Figure supplement 1**. COPI machinery localizes to the surface of LDs.

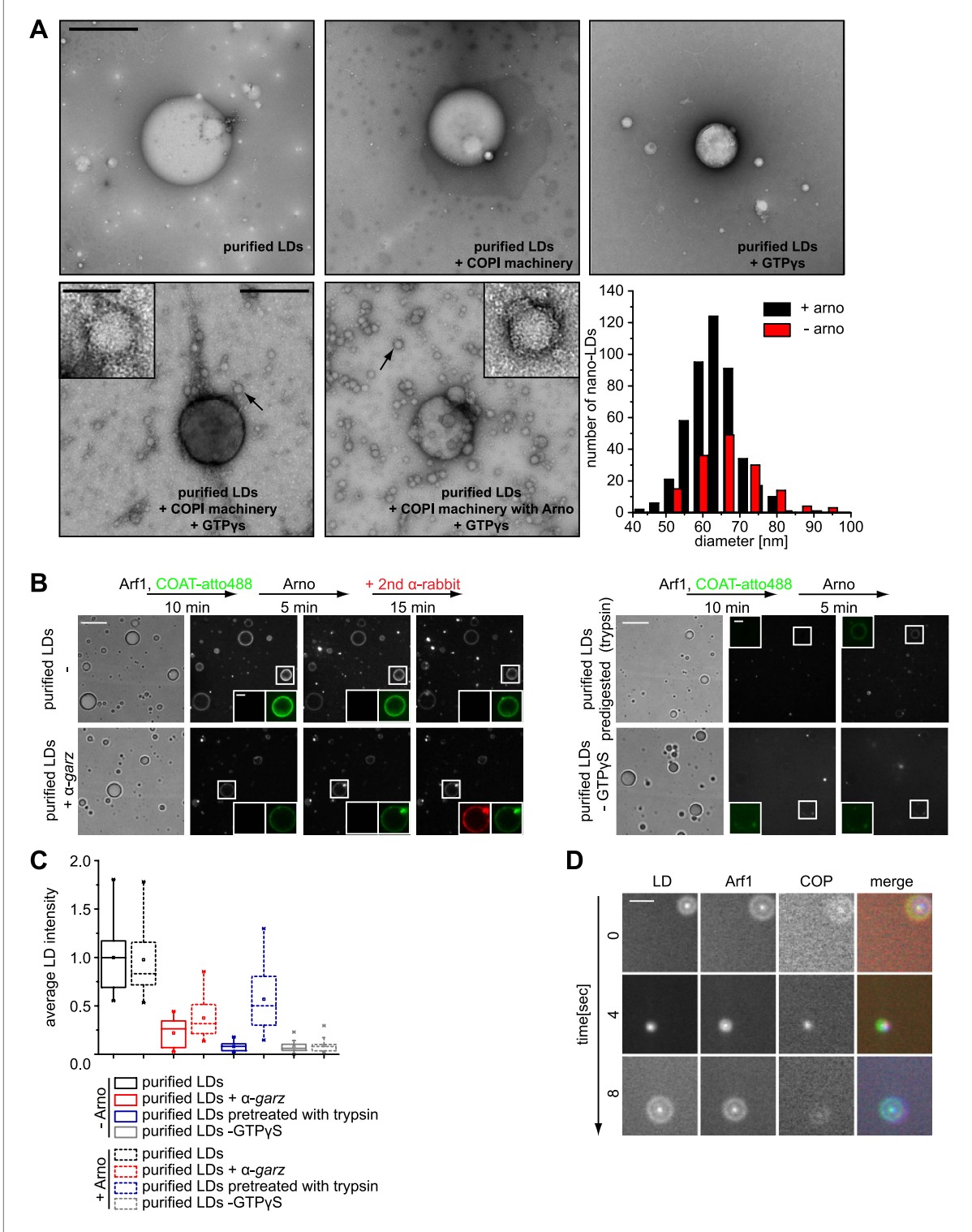

**Figure 4**. Arf1/COPI bud nano-LDs from purified, cellular LDs. (**A**) Purified LDs from S2 cells were incubated with components of the Arf1/COPI machinery in the presence or absence of GTPγS. Representative electron micrographs reveal abundant nano-LDs formed in the presence of activated Arf1/COPI. Scale bars are 500 nm (overview) or 100 nm (inset). Histograms show the size distribution of nano-LDs formed. (**B**) Purified LDs have the ability to activate Arf1 by GTP loading. Purified LDs incubated with Arf1, GTPγS, and fluorescently labeled COPI, but without the addition of a nucleotide exchange factor, are able to recruit COPI (green) in a GTP-dependent manner (top left panel and bottom right panel). COPI binding is abolished by blocking the exchange factor garz with an antibody (bottom left panel), or by digesting LD proteins with trypsin prior to the experiment
*Figure 4. Continued on next page*

*Figure 4. Continued*

(top right panel). Recruitment is partially restored by addition of a soluble Arf1-GEF, ARNO. Adding a secondary antibody (red) that recognizes the αgarz antibody labeled LDs dependent on the presence of the primary antibody. Scale bars are 5 µm (overview) or 1 µm (inlay). (**C**) Quantification of the recruitment of COPI to purified LDs. For each experiment in (**B**) the average intensity of 15 LDs was determined. (**D**) Nano-LDs formed from cellular LDs into the buffer visualized by fluorescence microscopy detecting Arf1 (red), COPI (green) and LDs (MDH labeled, blue). Scale bar is 5 µm.

The following figure supplements are available for figure 4:

**Figure supplement 1**. Purified LDs were incubated with Arf1-Y35A, coatomer, ARNO and GTPγS, upon budding conditions shown in *Figure 4A*.

performed with the Arf1Y35A mutant, which is deficient in dimer formation (*Beck et al., 2008*; *Figure 4—figure supplement 1A,B*). This lack of requirement for dimerization of Arf1 in nano-LD formation might reflect a lower energy barrier in the scission step of budding off a nano oil droplet compared with a vesicle.

We next tested whether purified LDs can activate Arf1 by GTP loading and, as a consequence, form COPI nano-LDs. Addition of ARNO, a soluble Arf1 nucleotide exchange factor to the in vitro budding reaction increased the number of nano-LDs observed, demonstrating that exchange activity was limiting (*Figure 4A*). Using fluorescently labeled coatomer, we observed recruitment to LDs in such reactions in a GTP-dependent manner (*Figure 4B*). COPI binding to LDs was abolished efficiently by blocking the exchange factor *garz* with an antibody or by digesting LD proteins with trypsin before the experiment (*Figure 4B,C*). In either case, recruitment could be partially restored by adding a soluble Arf1-GEF, ARNO. When we added a secondary antibody against the α*garz* antibody, LDs were labeled if the primary antibody was present, further indicating that Arf1-GEF was on the LDs (*Figure 4B*).

In addition to COPI labeling of the LD surface in these reactions, we observed nano-LDs (stained by BODIPY) in the supernatant from reactions containing fluorescently labeled Arf1 (Cy3) and COPI (Alexa647), as well as GTPγS (*Figure 4D*; *Video 2*), directly demonstrating nano-LD formation from isolated LDs.

## Modulating lipid droplet surface properties rescues lipid droplet protein targeting defects due to lack of Arf1/COPI

The budding of nano-LDs, with a very high surface to volume ratio, from the surface of donor LDs is predicted to remove primarily phospholipids. Therefore LDs from Arf1/COPI-depleted cells should contain more phospholipids than LDs from control cells. Indeed, when we compared lipids of purified LDs from cells depleted of βCOP with those from control cells, we found the levels of phosphatidylcholine (PC) and phosphatidyletha-nolamine (PE), but not TG, increased (*Figure 5A*).

We previously discovered that the enzyme CCT1, catalyzing the rate-limiting step of PC synthesis, binds to LDs deficient in PC, effectively acting as a biosensor for PC on expanding LDs (*Krahmer et al., 2011*). We therefore reasoned that Arf1/COPI depletion, by causing increased PC levels on LDs, would affect the time course of CCT1 recruitment to LDs. Indeed, CCT1 localized to LDs at later times during LD expansion (*Figure 5B*).

The model of Arf1/COPI removing primarily phospholipids from donor LDs predicts that the effects of Arf1/COPI depletion might be overcome

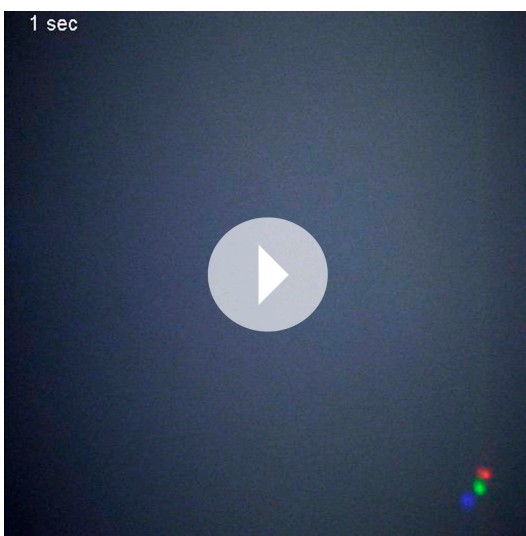

**Video 2**. Time lapse video of Arf1/COPI coated nano-LDs formed from cellular LDs. Nano-LDs are visualized by fluorescence microscopy detecting Arf1 (red), COPI (green) and LDs (MDH labeled, blue). Shifts between channels are due to short time delays between channel acquisitions, during which nano-LDs diffuse in solution.

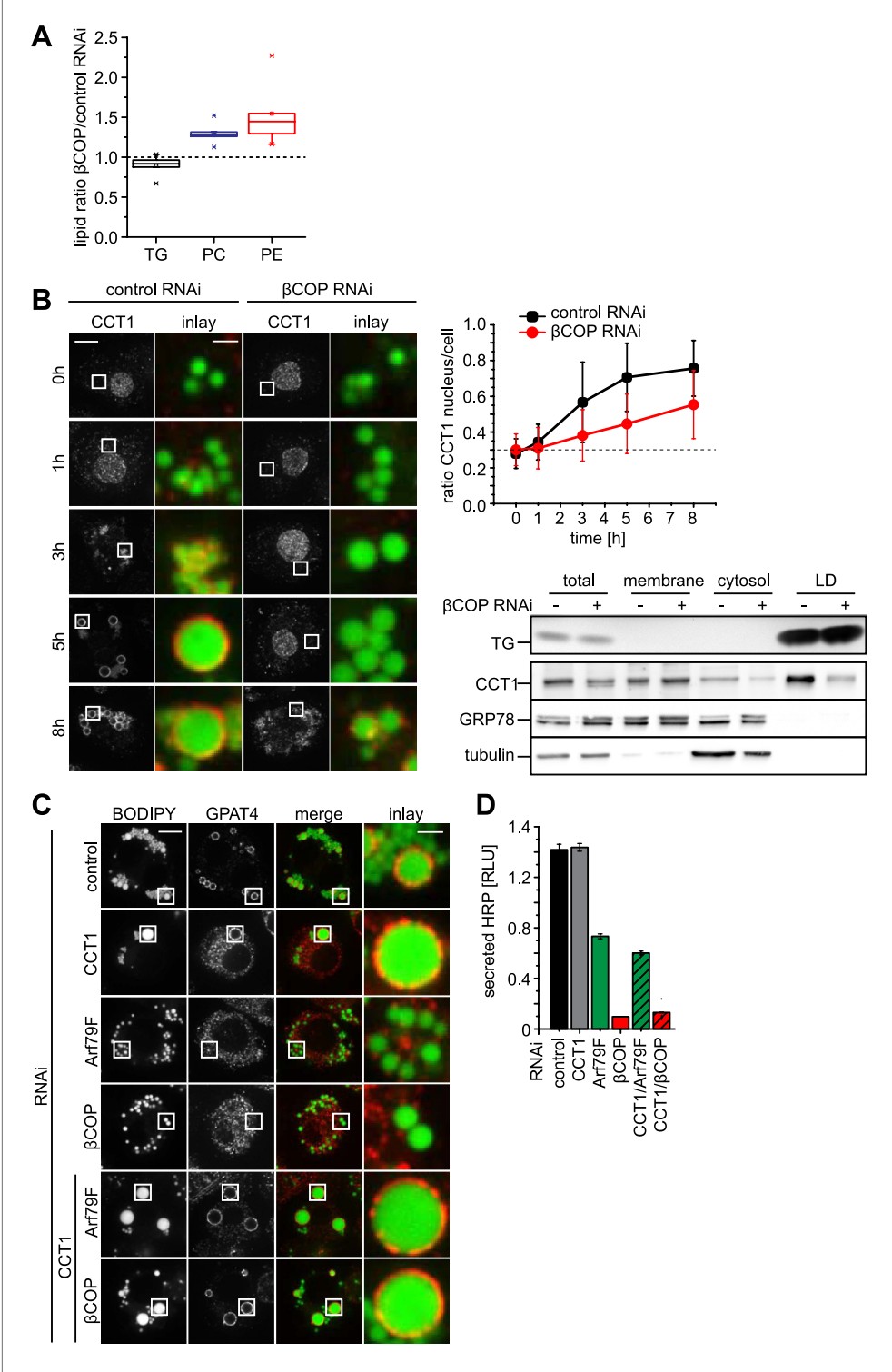

**Figure 5**. Lack of Arf1/COPI increases phospholipids on LDs, abolishing GPAT4 LD targeting. (**A**) PC and PE, but not TG levels are increased in LDs from βCOP depleted cells compared with WT cells. (**B**) LD (green) targeting of endogenous CCT1 (red) is delayed in cells depleted of βCOP. Time = 0 indicates the addition of oleate to the cells. Ratios between nuclear and LD targeted CCT1 signals are shown. Error bars represent the SD of the mean ratio from 100 cells. Western blot analysis shows decreased targeting of CCT1 to LDs when cells are depleted for βCOP. (**C**) Efficient co-depletion of CCT1 and Arf1 or βCOP restores GPAT4 targeting to LDs even in the absence of a

*Figure 5. Continued on next page*

*Figure 5. Continued*

functional COPI machinery. (**D**) Arf1/CCT1 or βCOP/CCT1 co-depletion blocks HRP secretion. Error bars represent the SD of triplicate measurements.

The following figure supplements are available for figure 5:

**Figure supplement 1**. Depletion of COPI machinery components is efficient.

by alternative treatments limiting the availability of phospholipids for LDs. To test this prediction, we decreased PC synthesis by depleting CCT1, either alone or in combination with Arf79F or βCOP. Depletion efficiency was equally efficient in single and double depletions (*Figure 5—figure supplement 1*). As expected from previous studies, CCT1 depletion resulted in coalescence of LDs into giant LDs, due to limiting availability of phospholipids on LDs (*Figure 5C*; *Guo et al., 2008*; *Krahmer et al., 2011*). Also, as predicted, CCT1 depletion did not abolish GPAT4 targeting to LDs. Strikingly, when CCT1 was depleted concomitantly with Arf79F or βCOP, GPAT4 targeting to LDs was efficiently restored (*Figure 5C*). The ability of CCT depletion to complement deficient Arf1/COPI function was specific to the GPAT4 targeting to LDs, as CCT1 depletion did not restore the defect in protein secretion due to Arf79F or βCOP depletion (*Figure 5D*).

If Arf1/COPI proteins function to remove phospholipids from LDs and thus allow membrane bridges to be established between the ER and LDs, then modulating the LD surface properties by other means should similarly alter protein targeting to LDs. To test this prediction, we added PC to cells. In agreement with the hypothesis, adding excess PC prevented GPAT4 targeting to the LD surface (*Figure 6A*). We suspect that, in this experiment, PC accumulates on LD surfaces and shields their TG cores, thereby lowering surface tension, and thus might prevent the establishment of membrane bridges with the ER.

Conversely, we predicted that a surfactant with a low potential to shield TG, therefore generating higher LD surface tension, might restore GPAT4 targeting to LDs even in the setting of Arf1/COPI depletion. We hypothesized that cholesterol (which in *Drosophila* cells is normally only present in very low amounts), with its small head-group and pronounced cone shape, might act in this manner. As predicted, in vitro measurements confirmed that cholesterol addition increased the surface tension of a TG-buffer interface when added in the presence of phospholipids (PC and PE) mimicking the LD surface composition (*Figure 6—figure supplement 1A*). Additionally, emulsion stability was reduced by cholesterol (*Figure 6—figure supplement 1B*). When cholesterol was added to Arf1/COPI-depleted cells, the cholesterol content increased at LDs (*Figure 6—figure supplement 1C,D*). Importantly, adding cholesterol to cells was sufficient to restore targeting of GPAT4 to LDs in Arf1/COPI-depleted cells (*Figure 6A*), and the number of GPAT4-positive LDs depended on the ratio of cholesterol and PC added to cells (*Figure 6A*). To test whether the effect is due to cholesterol's physical properties, or alternatively, to some physiological change in the cells induced by cholesterol, we repeated these experiments with SR59230A and stearylamine. Both of these agents are surface active, amphiphilic molecules that normally do not occur in cells, but which induce LD destabilization in vivo (*Murphy et al., 2010*), likely by increasing LD surface tension or by decreasing line tension of coalescence intermediates. In agreement with the findings with added cholesterol, adding SR59230A or stearylamine efficiently restored GPAT4 targeting to Arf1/COPI-depleted LDs (*Figure 6B*).

To further test whether changes of LD surface properties, introduced by the action of Arf1/COPI, controls GPAT4 targeting to LDs through membrane bridges, we reconstituted this reaction in vitro with a microfluidic device (*Figure 6—figure supplement 1E*). We introduced microsomes harboring GFP-GPAT4 into buffer-in-oil micro-reactors (*Figure 6C*). Mixing the content of the micro-reactors by flow through zig-zagging micro-channels led to localization of some GPAT4 to the monolayer delimiting the TG phase. The amount of GFP-GPAT4 at the monolayer depended on its composition and varied according to the surface tension. Similar to the situation in cells, monolayers rich in cholesterol and having higher surface tension, bound GFP-GPAT4 more efficiently than control monolayers of PC and PE (*Figure 6C*).

## Discussion

The Arf1/COPI machinery is important for governing LD morphology, protein targeting, and consequently lipolysis (*Beller et al., 2008*; *Guo et al., 2008*; *Soni et al., 2009*; *Ellong et al., 2011*). However,

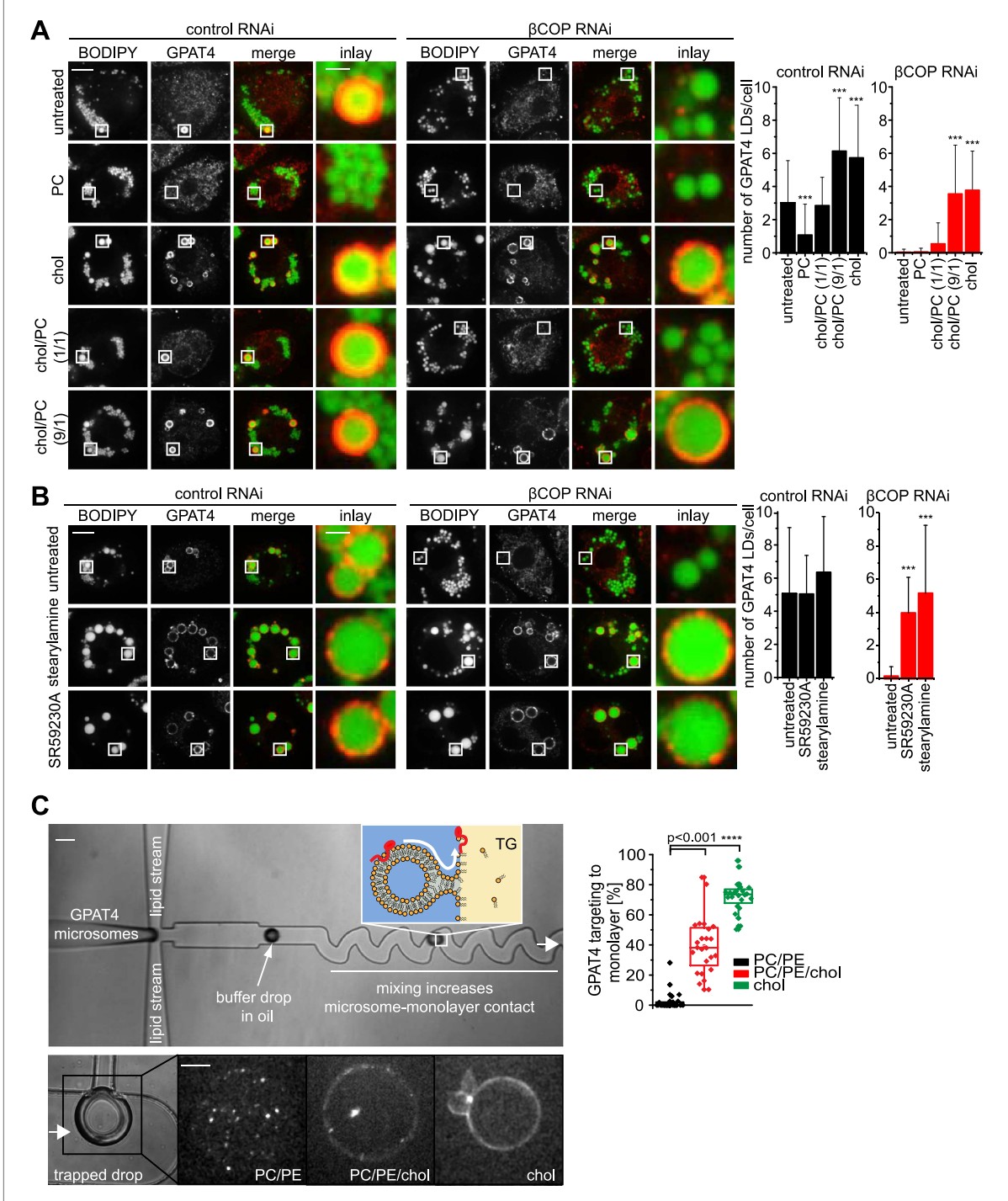

**Figure 6**. LD surface properties modulate GPAT4 LD targeting. (**A**) Addition of exogenous PC to S2 cells inhibited GPAT4 LD targeting in βCOP or control RNAi-treated cells. Cholesterol (chol) addition to cells restored GPAT4 LD targeting in βCOP-depleted cells. Targeting efficiency depends on the ratio of added cholesterol and PC in βCOP or control RNAi-treated cells. (**B**) The artificial compounds SR59230A or stearylamine rescued GPAT4 LD targeting in βCOP depleted cells. The numbers of GPAT4-targeted LDs per cell are shown. Error bars represent the SD from the mean number of GPAT4-targeted LDs in 40 cells. Statistical significance was calculated using ANOVA, followed by a Dunnett test with a 99% confidence interval (p=0.01). (**C**) GPAT4 targeting to phospholipid monolayers depends on the surface tension. Buffer drops containing GPAT4-GFP-labeled microsomes are formed in a microfluidics device by flow focusing. The buffer micro-reactors are surrounded by oil of different composition (TG containing PC/PE (0.25% ea.) or PC/PE (0.25% ea.) + 2% cholesterol, or cholesterol only (0.5%); concentrations are w/w compared to TG). Each formed buffer drop pass through a zigzag

*Figure 6. Continued on next page*

*Figure 6. Continued*

region where microsomes inside the buffer drop are constantly brought into contact with the monolayer at the oil interface. Drops are arrested in a network of trapping chambers. In the presence of PC/PE, little GPAT4-GFP is targeted to the monolayer but stays in microsomes. Addition of 2% cholesterol or cholesterol alone significantly increased GPAT4-GFP signal on the monolayer. Quantification of the relocalization efficiency of GPAT4 from microsomes to the monolayer interface. Bar = 100 µm (device) or 25 µm (drop).

The following figure supplements are available for figure 6:

**Figure supplement 1**. Cholesterol leads to an increase of surface tension at a TG/buffer interface.

how Arf1/COPI proteins act to affect LDs has been unknown. Recent evidence from in vitro experiments using artificially generated oil-water interfaces show that GTP-bound Arf1 and COPI proteins are sufficient to bud nano-LDs (*Thiam et al., 2013a*), suggesting that the Arf1/COPI machinery might perform a similar function at the oil-water interfaces of LDs in cells.

The current studies show that Arf1/COPI machinery has an additional function other than its canonical function in forming bilayer vesicles namely that these proteins can control the formation of membrane bridges between LDs and the ER to mediate targeting of specific proteins (such as ATGL, GPAT4, and DGAT2) from the ER to LDs.

Taken together, our data are most consistent with a model for the function of Arf1/COPI in which these proteins act directly on LDs to remove phospholipids from the LD surfaces through the formation of nano-LDs. Budding of nano-LDs in turn, increases surface tension of the donor LD and allows membrane bridges to be established between this LD and the ER. These membrane bridges provide a pathway for the localization of membrane-associated proteins, such as ATGL and GPAT4, and it allows them to diffuse to the LD surface where they perform key steps in TG metabolism. Without functional Arf1/COPI, TG synthesis enzymes fail to target LDs, which as a consequence cannot expand to form large LDs. In addition, as reported (*Beller et al., 2008*; *Soni et al., 2009*), ATGL fails to target LDs leading to a defect in lipolysis and a mild increase in the size of small LDs. Consistent with this model, incubation of cells depleted for components the Arf1/COPI machinery with chemicals that increase LD surface tension, such as cholesterol, stearylamine or SR59320A is sufficient to restore GPAT4 targeting. Various proteomic, biochemical, and cell biological studies showing that components of the Arf1/COPI machinery are present on LDs (*Nakamura et al., 2005*; *Bartz et al., 2007*; *Guo et al., 2008*; *Ellong et al., 2011*; *Bouvet et al., 2013*) are also consistent with this model.

Calculations based on the size of the targeted LDs and the formed nano-LDs suggest that only a few nano-LD budding events are required to significantly increase the surface tension of the donor LD (*Thiam et al., 2013a*). Thus, Arf1/COPI activity that results in the budding of nano-LDs will cause a significant change in the surface properties of existing LDs, and these changes are required to enable interactions of the LD monolayer surface with bilayer membranes. Specifically, we posit that the increase in the surface tension of LDs allows for the formation of bridges with the ER, whereas the densely packed phospholipid shell on LDs with low surface tension, where Arf1/COPI have not acted, are refractive to forming a bridge with the ER.

In an alternative and possibly complementary model, Arf1/COPI might also function to maintain ER lipid composition or structure in a manner that allows for the formation of bridges with LDs. In other systems inhibition of the Arf1 guanine-nucleotide exchange factor led to collapse of the Golgi apparatus into the ER and ectopic cleavage and activation of the transcription factor SREBP (*Walker et al., 2011*). In *Drosophila,* SREBP up-regulates phospholipid synthesis (*Dobrosotskaya et al., 2002*), which could indirectly affect LD surface properties (*Krahmer et al., 2011*). However, in our experimental system, we did not detect increased SREBP cleavage, up-regulation of the SREBP target genes (such as CCT1, acetyl-CoA synthase, acetyl-CoA carboxylase and fatty acid synthase) or changes in cellular PC or PE levels after Arf1 depletion (*Figure 5—figure supplement 1* and MJO and TCW, unpublished observations).

Once ER-LD bridges are established, GPAT4, and presumably other enzymes (e.g., AGPAT3, DGAT2, or ATGL/*brummer*) rapidly migrate to LDs. The time course of enzyme relocalization, in our experiments triggered at some point during oleate loading or after adding back COPI by cell–cell fusion, suggests that, once LD-to-ER bridges are established, targeting is limited by diffusion across the bridges. It is unclear how cargo that migrates from the ER to LDs is selected. Intriguingly, the Arf1/COPI

mechanism appears to operate specifically for proteins that are embedded in the membrane, such as GPAT4 and ATGL, which behaves similarly to GPAT4 as an integral membrane protein (*Soni et al., 2009*, and FW, MJO, RVF, and TCW, unpublished observations). It is also unclear why these LD-targeted proteins accumulate on LD surfaces. Accumulation could be mediated by partitioning into the oil phase, but the mechanism providing energy for the reaction is not yet known.

Our findings provide a number of new questions for investigation. It is unknown if the Arf1/COPI machinery is constitutively active stochastically on some LDs or if is regulated. Interestingly, data from in vitro budding reactions from oil-water interfaces indicate that Arf1/COPI can act only on membranes sufficiently covered by phospholipids (*Thiam et al., 2013a*). Therefore, Arf1/COPI might be part of a system that detects LDs that are sufficiently coated by PC (i.e., have reached a sufficiently low surface tension) and thus are suitable for further expansion. It is also unclear how Arf1/COPI-mediated protein targeting is affected by lipolysis. Generation of surface-active lipids during lipolysis, such as fatty acids or diacylglycerol, might increase LD surface tension and subsequently augment the triggering of ER-LD bridge formation, thereby allowing more lipases to migrate to LDs. Also unclear is how the specificity of membrane bridge formation of LDs to the ER is controlled. Finally, it will be of interest to determine the fate of the nano-LDs formed by Arf1/COPI actions. Nano-LDs are similar in size to typical COPI vesicles. However, in contrast to vesicles, they are made up of a small oil core that is likely coated with a monolayer of phospholipids and may contain specific proteins.

The model emerging from our studies highlights how cells solved a fundamental problem—how to deal with LDs, which are essentially emulsified oils in the aqueous cytosol. Through the actions of the Arf1/COPI machinery, the surface properties of LDs can be altered such that proteins are able to access them. Among all coat complexes known to function in vesicular trafficking, the Arf1/COPI system has unique properties that make it ideally suited to function in this process. All other vesicular coat complexes require exchange factors that contain trans-membrane spanning protein segments, which are unlikely to be found on LDs. Arf1/COPI does not require such a factor. By this unique mechanism, cells can alter the surface properties of LD emulsions and enable them to interact with membranes, so that specific enzymes can gain access to LDs and facilitate dynamic changes in lipid storage or utilization. Our findings additionally provide evidence for a previously unrecognized cellular mechanism by which Arf1/COPI proteins can control protein trafficking.

## Materials and methods

### Antibodies
Rabbit polyclonal antibodies used: anti-GPAT4 (*Wilfling et al., 2013*), anti-CCT1 (*Wilfling et al., 2013*), anti-GBF1 (BD Biosciences, San Jose, CA), anti-KDEL-receptor (KDELR; gift from Dr JE Rothman; Yale University), anti-βCOP (gift from Dr JE Rothman; Yale University), anti-perilipin3 (TIP47; Novus Biologicals, Littleton, CO), anti-αCOP (Abcam, Cambridge, MA), anti-GRP78/BiP (ET-21) (Sigma–Aldrich, St. Louis, MO) and anti-garz (*Wang et al., 2012*) (gift from Dr A Paululat; University of Osnabrück). Mouse monoclonal antibodies used: anti-GM130 (BD Biosciences), anti-tubulin (Sigma–Aldrich), anti-β'COP (gift from Dr JE Rothman; Yale University), and anti-clathrin heavy chain (x22) (Thermo Scientific, Waltham, MA) antibody. The following secondary antibodies were used: Alexa Fluor 568 goat anti-rabbit (Invitrogen, Grand Island, NY), Alexa Fluor 488 goat anti-mouse (Invitrogen), ATTO 647N (STED) goat anti-rabbit (Active Motif, Carlsbad, CA), and goat anti-mouse STAR470SX (Abberior, Göttingen, Germany).

### Plasmid DNA construction
Full-length cDNA encoding CG5295 (*brummer*) and CG10374 (Lsd1) were obtained from the DGRC (https://dgrc.cgb.indiana.edu/) and subcloned into the pENTR/SD/DTOPO vector (Invitrogen) and indicated destination expression vectors (actin promoter). The destination vectors used in this study are part of the *Drosophila* Gateway Vector Collection and are available from the DGRC (https://dgrc.cgb.indiana.edu/).

### Cell culture and transfection
WT *Drosophila* S2 or stably transfected cells (pAGW-*brummer* or pA*Cherry*W-Lsd1) were cultured, treated with oleate, transfected and depleted by RNAi as described (*Krahmer et al., 2011*). Cells were analyzed 4 days after RNAi treatment. *Table 1* contains a list of primers to generate dsRNAs for RNAi. A segment of pBluescript backbone was used as the template for control RNAi. Expression of the ss-HRP construct was induced and the secretion assay was performed as previously described (*Bard et al., 2006*). If not otherwise indicated, cells were treated after RNAi treatment with 1 mM oleate for 8 hr.

**Table 1.** Sequences of primers used for RNAi experiments

| Gene | Gene ID | Forward | Reverse |
|------|---------|---------|---------|
| garz | CG8487 | TTGCACAAACTTTGATTCCTG | CATATCGGCGCACTATAATC |
| Arf79F | CG8385 | TAGCGATTAGCGTTCTTCA | CTGCCAAATGCAATGAACG |
| αCOP | CG7961 | AGGAAGCTAAGCTTGTCAAA | GGACGAGTCTGGAGTGTTT |
| βCOP | CG6223 | CCAGTCAGTTGGGTGACCTT | CCTAGCAAGCCCATAACCAA |
| β'COP | CG6699 | ATCTTGCTTCCCACAACGTC | CCGAAGGACAACAACACCTT |
| γCOP | CG1528 | ATTACGTTCACAGCACGCAG | CAGAGGAGGGCTATGACGAC |
| ζCOP | CG3948 | CCGTCGCAGATCTCGTC | GCATCCTGGCCAAGTACTA |
| εCOP | CG9543 | AGGTGCCAGATGTTGGTCTC | CCAACTCGGTGCTATTCGAT |
| δCOP | CG14813 | AAGCTGTCTGCGCCATAAAT | TCCAGTGGCACATTCCAATA |
| CCT1 | CG1049 | ACATCTATGCTCCT1CTCAAGGC | CTCTGCAGACTCTGGTAACTGC |
| pBluescript | | AATTCGATATCAAGCTTATCGAT | TAAATTGTAAGCGTTAATATTTTG |

Exogenous lipids (PC, or cholesterol, or PC/cholesterol) were added to *Drosophila* S2 cells at the second day of the RNAi treatment. The final concentration of these lipids in the growth medium was 5 mM. On the fourth day, medium was replaced by fresh medium containing 1 mM oleate and LD formation was induced for 8 hr before cells were fixed. The artificial lipid SR59230A was added to RNAi treated cell during the last hour of oleate treatment to a final concentration of 100 μM. NRK cells were cultured in DMEM with 10% FBS and antibiotics (100 units of penicillin and 100 μg of streptomycin per ml). Cells were split onto glass bottom plates and incubated in the culture media the day before imaging. LDs were induced by treatment with 0.5 mM oleate for 2 hr before fixation and imaging.

## Protein purification

Fluorescently labeled Arf1 was generated using an Arf1-variant in which the single cysteine residue of Arf1 was replaced with serine, and the C-terminal lysine was replaced with cysteine, yielding Arf1-C159S-K181C. Published work has demonstrated that exchanging the C-terminal lysine of the small GTPase with a Cys- residue, and subsequent fluorescent labeling (using thiol-reactive dyes on Cys181), does not inhibit Arf1-function (*Beck et al., 2008*; *Manneville et al., 2008*). In short, human Arf1-C159S-K181C and yeast N-myristoyltransferase were coexpressed in *Escherichia coli* supplied with BSA-loaded myristate. Cell lysates were subjected to 35% ammonium sulfate, and the precipitate, enriched in myristoylated Arf1, was further purified by DEAE-ion exchange. Eluted fractions of interest were concentrated in spin-column filters with a 10-kD cutoff (Millipore), and fluorescently labeled using Cy3-maleimide (GE Healtcare) according to the manufacturer's protocol. To remove excess dye, samples were purified by gel filtration using a Superdex 75 column (GE Healthcare).

Recombinant coatomer protein was expressed and purified, as described in *Sahlmuller et al., (2011)*. In short, Sf9 insect cells were infected with baculovirus encoding for heptameric coatomer. Coatomer complexes were isolated from the soluble protein fraction by nickel-affinity purification, concentrated in spin-column filters with a 250-kD cutoff (Millipore), and fluorescently labeled using Alexa-Fluor-647-NHS (Molecular Probes) according to the manufacturer's protocol. Excess imidazole and dye was removed by gel filtration using a Superose 6 column (GE Healthcare).

## Lipid droplet size measurements

Cells were treated with 1 mM oleate, stained with BODIPY, and subsequently imaged and measured as described (*Wilfling et al., 2013*). Density plots were computed using R (http://www.r-project.org/).

## Light microscopy

For live-cell imaging and immunostaining, cells were prepared and imaged as described (*Wilfling et al., 2013*). The antibody dilution buffer used for immunostaining of perilipin3 in NRK cells did not contain detergent. The permeabilization buffer used for immunostaining of CCT1 in *Drosophila* S2 cells had a final concentration of 0.1% NP-40. Also, the buffer for first and secondary antibody dilution

was detergent free. LDs were stained with 1 µg/ml BODIPY (Invitrogen) or LipidTOX (Invitrogen) or 10 mM of MDH (*Yang et al., 2012b*).

## STED microscopy

STED microscopy (*Hell and Wichmann, 1994*) was performed on a custom-built system featuring an 80 MHz mode-locked Ti:Sapphire laser (Chameleon Ultra II, Coherent) tuned to either 760 nm or 770 nm as the depletion beam. The 140 fs pulses output from this laser were stretched to several hundred picoseconds using a glass block and a 100 m polarization-maintaining optical fiber (Thorlabs) to prevent multiphoton excitation of the fluorophores. A spatial light modulator in the depletion beam path allowed phase modulation for generating a toroidal depletion focus in the sample and for correction of system induced optical aberrations (*Gould et al., 2012*). For fluorescence excitation, 510 nm and 640 nm pulsed diode lasers (PicoQuant) were electronically synchronized to the depletion beam and an electronic delay (Colby Instruments) allowed adjustment of the relative arrival time of the laser pulses at the sample. Excitation and STED beams were combined using dichroic mirrors and focused into the sample through a 100×/1.4NA oil immersion objective lens (UPLSAPO 100XO/PSF, Olympus). Imaging was preformed via beam scanning. A 16 kHz resonant scanner and a galvanometer mirror (EOPC) were placed in the beam path and imaged into the pupil plane of the objective lens to scan the beams through the sample. Fluorescence from the sample was collected by the objective lens, de-scanned by the scan mirrors, separated from laser light using dichroic mirrors, bandpass filtered (FF01-685/40 for ATTO647N or FF01-593/46 for STAR470SX; both from Semrock), and focused onto 105 µm core diameter (ATTO647N: ~0.7 Airy units; STAR470SX: ~0.8 Airy units) multimode fibers (Thorlabs) connected to single-photon counting avalanche photodiodes (APD; ARQ-13-FC, Perkin Elmer). APD counts were acquired using a FPGA-based data collection board (PCIe-7852R, National Instruments) and custom acquisition software programmed in LabView (National Instruments). Recorded pixel values were linearized (on the DAQ card) to account for the sinusoidal velocity profile of the resonant mirror and normalized according to the pixel dwell times such that the center pixel was divided by unity. Dual-color imaging of ATTO647N and STAR470SX were performed using sequential frame acqustions similar to previously published reports using a long Stoke's shift fluorophore as the second color channel (*Schmidt et al., 2008*). Laser powers (measured at the objective back aperture) were ~16 µW of 510 nm excitation light and ~130 mW of 760 nm STED light for STAR470SX and ~17 µW of 640 nm excitation light and ~130 mW of 770 nm STED light for ATTO647N. Images were acquired with a 20 nm pixel size in a 1024 by 1024 image format with 500 accumulations per line, resulting in a frame rate of 0.032 Hz.

## Comparison of colocalization between the experiment and a random distribution

To assess whether the overlapping signals of βCOP, clathrin, KDELR (in NRK cells) and αCOP, garz (for S2 cell) with BODIPY was erratic a Matlab script was written.

The population of the immunostained foci was denoted **A** and BODIPY stained LDs were denoted **LD**. For each colocalization experiment, a minimum of 15 snapshots was taken. Each image was first analyzed to assess the frequency of colocalization between **A** and **LD** from the experiment (1)**,** from a random situation where **A**-type particles were randomly distributed (2); both situations were then compared (3).

1. The brightness contrast was adjusted for each channel of the picture using ImageJ. After applying a threshold binary images were generated for each channel. The total number of A-type particles, nA, and their corresponding radius, rA, were determined. For each LD particle, the distance of the first A-type neighbor was determined. Negative distances corresponded to overlapping of A and LD. The colocalized fraction of A-type particles with LD population was given by nexp/nA, where nexp was the number of colocalized A-type spots (number of negative distances).

2. The random distribution of A particles was based on an analytical model following a binomial distribution hypothesis. The choice of a binomial distribution model was adequate to assess over-dispersion of A-type particles (*Rosner, 2011*). From the binary mask of the LD population, each LD of radius rLD was given a new radius rLD + rA. The probability of colocalization of A and LD can be formulated by the probability of having an A-type dot colocalizing to a LD of the new defined radius. In the total field occupied by the cells (areaf denoting for the area of the field), the total area fraction occupied by LD is given by:

$$s = \frac{\sum_{LD} \pi(\mathbf{rLD} + \mathbf{rA})^2}{\mathbf{areaf}}$$

The probability to colocalize **n** A-type dots out of **nA** is:

$$P(n) = \subset_{nA}^{n} \; s^n (1 - s)^{nA - n}$$

Colocalization of $n_s$ ($n_s = s*nA$) from the **nA** particles has the highest occurrence (**P(n)** < **P(n$_s$)**). Therefore the most likely situation from a random distribution was the colocalization of $n_s$ A-type particles with **LD**. Likewise we observed that a random simulation based on a normal distribution results in similar values for $n_s$ (data not shown).

3. If $n_s \gg n_{exp}$, particles **A** are excluded from LDs; if $n_s \ll n_{exp}$, they are enriched on LDs.

## Quantitative real-time PCR
Expression levels were measured by quantitative Real-Time PCR. Total RNA was prepared with the RNeasy Mini Kit (Qiagen); 1 µg was used for first-stand cDNA synthesis with the iscript cDNA synthesis kit (BioRad). Real-time quantitative PCR was performed on a LightCycler 480II (BioRad) using the Power SYBR green mix (Applied Biosystems). Pimers used are listed in *Table 2*.

## TLC and lipid measurements
Purfication of lipid droplets was done as previously reported (*Wilfling et al., 2013*). Lipids were extracted (*Folch et al., 1957*), separated on silica TLC plates (Merck) with chloroform/methanol/acetic acid/formic acid/water (vol/vol) (70:30:12:4:1) for phospholipids or n-heptane/isopropyl ether/acetic acid (60:40:4) for neutral lipids, and detected by Hanessian's

## Electron microscopy
Purified LDs incubated with ARF/COPI at various conditions were absorbed to continuous carbon-coated grids (glow discharged) at room temperature for 5 min, rinsed briefly with HKM buffer (25 mM HEPES-KOH at pH 7.4, 100 mM KCl, 10 mM MgCl$_2$), and stained with 1% uranyl formate for 20 s. Negatively stained samples were imaged under low-dose conditions in an FEI Tecnai12 microscope (120 kV). Micrographs were collected at 26,000 × magnification using Gatan 4K × 4K CCD camera, giving a pixel size of 4.5 Å. The diameters of nano LDs were manually measured on digital micrographs.

## Surface tension measurements
The surface tension of different lipids or lipid mixtures was measured using a drop weight method. HKM buffer containing different concentrations of phospholipids and/or cholesterol was formed in a TG oil phase. Buffer drops were slowly formed in the oil (at a flow rate of 20 µl/hr) to allow dynamic interfacial equilibrium. At a critical size the drop detaches. For each concentration, videos of this process were taken using a 1394 Unibrain camera. From the inner diameter *d* of the injection tube (d = 250 µm), the surface tension is determined by *mg/(π*d*f)* where *f* is a Wilkinson geometric parameter correction that depends on the ratio between *d* and the radius of the detached drop and *g* is the gravity constant. The mass *m* of the drop was calculated according to *m = v*Δρ* (*v* is the volume of the drop and *Δρ* is the volume mass difference between oil and buffer phases).

**Table 2.** Sequences of primers used for RT-PCR

| Gene | Gene ID | Forward | Reverse |
|------|---------|---------|---------|
| GAPDH1 | CG12055 | TTGTGGATCTTACCGTCCG | ACCTTAGCCTTGATTTCGTC |
| Arf79F | CG8385 | TTACAGTGTGGGATGTGGG | GAAGATAAGACCTTGTGTATTCTGG |
| βCOP | CG6223 | GACTTCTGCAATATCAAGGCC | GGTTTCGTAAACAATATTGCCG |
| CCT1 | CG1049 | GATACGGAGTGCGTCA | AATTCATCGGACAGAGTCCA |

## Stability assay of oil microdroplets in aqueous solution

2.5 mg of DOPE and 2.5 mg of DOPC (Avanti Polar Lipids) were solubilized in 250 mg TG (Sigma–Aldrich) by sonication. Lipids were then mixed with buffer (25 mM HEPES-KOH at pH 7.4, 100 mM KCl, 10 mM MgCl$_2$) in a ratio of 1/16 (oil/buffer) by vortexing and sonication for 5 min using a Branson 3510 sonicator water bath. The emulsion was added to the indicated amounts of cholesterol and sonicated for 2 min. The optical density of the emulsion was monitored over a time course of 2 hr in 1-min intervals by a TECAN infinite M200.

## Cell–cell fusion assay

*Drosophila* S2 cells were co-transfected with pAW-VSVG and pAW-*cherry*. After 24 hr cells were mixed 1:1 with a stable transfected cell line of GFP-GPAT4 depleted for β-COP for 4 days and treated for 8h with 1 mM oleate. The cell mixture was prepared for live cell imaging as described (*Wilfling et al., 2013*). Fusion of cells was initiated by addition of a low pH buffer (10 mM Na$_2$HPO$_4$, 10 mM NaH$_2$PO$_4$, 150 mM NaCl, 10 mM MES, 10 mM HEPES, pH 5) for 30 s. After incubation with the fusion buffer cells were immediately shifted to regular growth medium.

## Microfluidic experiments

The microfluidics device was fabricated by well-established soft lithography techniques. A wafer mold was made by lithography with a negative resin (SU8-2035). The device was made of a poly-dimethylsiloxane polymer, used to replicate the pattern on the mold and stuck on a glass cover slip. The height of the device is 58 ± 5 µm. A buffer and oil stream was generated using a syringe pump. By flow focusing, defined buffer drops were generated in the oil stream. These buffer micro-reactors contained GFP-GPAT4 microsomes. The oil used was a mixture of TG with phospholipids and/or cholesterol (PC/PE each 0.25% (wt/wt); PC/PE/cholesterol 1/1/4 with 2% (wt/wt) cholesterol; cholesterol 0.5% (wt/wt); indicated lipid concentrations are compared to TG). To ensure the same frequency of interaction of the microsomes with the monolayer interface, the flow rate of the buffer and oil stream (150 and 30 µl/hr) was kept constant for all experiments.

## Acknowledgements

We thank Drs Christopher Burd, Nora Kory, Natalie Krahmer, Gregory Lavieu, Lena Schroeder and Yuan Xue for critical discussions and Gary Howard for editorial assistance. We also thank Nora Kory, Huajin Wang and Yuan Xue for help with experiments. We thank Dr Vivek Malhotra for the HRP construct, Dr James Rothman for the βCOP, β'COP, and KDELR antibodies and Dr Achim Paululat for the *garz* antibody.

## Additional information

### Competing interests

Jörg Bewersdorf discloses significant financial interest in Vutara, Inc. The other authors declare that no competing interests exist.

### Funding

| Funder | Grant reference number | Author |
|---|---|---|
| National Institutes of Health | R01GM097194 | Tobias C Walther |
| National Institutes of Health | R01GM099844 | Robert V Farese Jr |
| Wellcome Trust | 095927/A/11/Z | Jörg Bewersdorf |
| Marie Curie BFLDs | | Abdou Rachid Thiam |
| National Institutes of Health | F32GM096859 | Travis J Gould |
| Boehringer Ingelheim Fonds | | Florian Wilfling |
| The Gladstone Institutes | | Robert V Farese Jr |
| The G. Harold and Leila Y. Mathers Foundation | | Tobias C Walther |

The funders had no role in study design, data collection and interpretation, or the decision to submit the work for publication.

## Author contributions

FW, ART, TCW, Conception and design, Acquisition of data, Analysis and interpretation of data, Drafting or revising the article, Contributed unpublished essential data or reagents; M-JO, Conception and design, Acquisition of data, Analysis and interpretation of data, Drafting or revising the article; JW, RB, TJG, ESA, Conception and design, Acquisition of data, Analysis and interpretation of data; FP, JB, Conception and design, Analysis and interpretation of data; RVF, Conception and design, Analysis and interpretation of data, Drafting or revising the article

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
