## [Decision Letter]

Thank you for sending your work entitled “Arf1/COPI Machinery Acts Directly on Lipid Droplet Surfaces to Enable Lipid Droplet Protein Targeting” for consideration at *eLife*. Your article has been favorably evaluated by a Senior editor and 3 reviewers, one of whom is a member of our Board of Reviewing Editors.

The Reviewing editor and the other reviewers discussed their comments before we reached this decision, and the Reviewing editor has assembled the following comments to help you prepare a revised submission. We feel the following are essential changes that must be included in a revised manuscript:

1) Show the effect of Arf1/COP1 loss on the trafficking of another enzyme, preferably of the triglyceride synthesis pathway, to the LDs.

2) Is Arf1 working as a monomer or dimer? The biogenesis of COPI at the Golgi membranes is reported to require dimeric Arf1. Either way this data will help reveal the role of Arf/COPI in vesicle formation.

3) Quantitate epsilon COP depletion.

4) Improve the quality of Figures 2 and 4. Figure 2—figure supplement 1: this is supposed to be a mathematical model describing the number of connections between the ER and a LD. To us, the panel on the left side looks like actual data, while the cartoon on the right references parameters (R,J,L) that are not described in the Figure legend or in the text of the manuscript. These are buried in the Methods, which is fine, but if the authors were hoping to convey some information through this figure they have failed. Also, it is unclear how the expected number of ER-LD connections (5-9) is arrived at from the Figure or the legend. Perhaps a bit more information in the figure legend would be helpful to explain the model in simple terms to the reviewers (and like most readers of *eLife*, we presume) who are not mathematicians.

*The images in Figure 4 fail to demonstrate colocalization. Video 2 does convince us that there is colocalization of all three markers on these fast moving “nano-LDs”, yet the way the imaging is performed, the still images are not useful. Either the authors need to figure out a way to image nano-LDs that are not moving (can they be collected and then pelleted onto a coverslip?), or perform the image acquisition with a much shorter time between channels, or else remove Figure 4 from the main figures (perhaps move it to the supplement, and definitely keep the video)*.

5) Write clearly in the Abstract, Introduction, and the Discussion previously published work on the role of COPI in budding from the LDs. Please also explain why the current findings are significantly more important compared with the published data.

---

## [Author Response]

*1) Show the effect of Arf1/COP1 loss on the trafficking of another enzyme, preferably of the triglyceride synthesis pathway, to the LDs*.

We now show that DGAT2 in addition to ATGL and GPAT4 requires the Arf1/COPI machinery for targeting Lipid Droplets (Figure 1—figure supplement 1).

*2) Is Arf1 working as a monomer or dimer? The biogenesis of COPI at the Golgi membranes is reported to require dimeric Arf1. Either way this data will help reveal the role of Arf/COPI in vesicle formation*.

We have added a number of experiments to address this question. First, we have repeated our in vitro budding reactions with a mutant form of Arf1 that impairs dimerization (Arf1Y35A; see Beck et al., JCB 2011). We find no difference in the formation of nanoLDs (Figure 4—figure supplement 1).

We also attempted a series of experiments to determine whether dimerization deficient mutants can rescue knockdown of endogenous Arf1. Unfortunately, these experiments were not productive due to a large variation of expression levels of even the wildtype proteins that did not consistently rescue the phenotype observed in all cells (presumably as too much to little of Arf1 was expressed from the cDNA).

*3) Quantitate epsilon COP depletion*.

These data are now included as Figure 1—figure supplement 1.

*4) Improve the quality of*
Figures 2 and 4*.*
*Supplemental figure 2A**: this is supposed to be a mathematical model describing the number of connections between the ER and a LD. To us, the panel on the left side looks like actual data, while the cartoon on the right references parameters (R,J,L) that are not described in the Figure legend or in the text of the manuscript. These are buried in the Methods, which is fine, but if the authors were hoping to convey some information through this figure they have failed. Also, it is unclear how the expected number of ER-LD connections (5-9) is arrived at from the Figure or the legend. Perhaps a bit more information in the figure legend would be helpful to explain the model in simple terms to the reviewers (and like most readers of eLife, we presume) who are not mathematicians*.

We agree with the referees. Initially, we had planned to have a figure supplement with more detail and straightforward presentation of the model. As an alternative, we have now moved most of the model section from “Materials and methods” to the figure legend and hope this will help readers. We have also expanded the explanation there to clarify the model.

*The images in Figure 4 fail to demonstrate colocalization. Video 2 does convince us that there is colocalization of all three markers on these fast moving “nano-LDs”, yet the way the imaging is performed, the still images are not useful. Either the authors need to figure out a way to image nano-LDs that are not moving (can they be collected and then pelleted onto a coverslip?), or perform the image acquisition with a much shorter time between channels, or else remove Figure 4 from the main figures (perhaps move it to the supplement, and definitely keep the video)*.

We now provide images with much shorter time interval that more clearly reveals the co-localization of Arf1, coatamer, and neutral lipid.

*5) Write clearly in the Abstract, Introduction, and the Discussion previously published work on the role of COPI in budding from the LDs. Please also explain why the current findings are significantly more important compared with the published data*.

As suggested, we have modified the Abstract, Introduction, and Discussion to better explain the significance and novelty of our findings and to integrate them better into the existing knowledge.